# Last Resort? Rationale for Comprehensive Molecular Analysis in Treatment-Refractory R/M HNSCC: A Case Report of Remarkable Response to Sacituzumab Govitecan Following Molecular and Functional Characterization

**DOI:** 10.3390/biomedicines13051266

**Published:** 2025-05-21

**Authors:** Henrike Barbara Zech, Philippe Schafhausen, Leonie Ramke, Janna-Lisa Velthaus, Simon Kreutzfeldt, Daniel Hübschmann, Kai Rothkamm, Carsten Bokemeyer, Anna Sophie Hoffmann, Stefan Fröhling, Hanno Glimm, Christian Stephan Betz, Malte Kriegs, Maximilian Christopeit

**Affiliations:** 1Department of Otorhinolaryngology, University Medical Center Hamburg-Eppendorf, 20251 Hamburg, Germany; 2Mildred-Scheel Cancer Career Center HaTriCS4, University Medical Center Hamburg-Eppendorf, 20251 Hamburg, Germany; 32nd Department of Medicine (Oncology, Hematology, Bone Marrow Transplantation with Section Pneumology) and University Cancer Center Hamburg (UCCH), University Medical Center Hamburg-Eppendorf, 20251 Hamburg, Germany; 4Division of Translational Medical Oncology, German Cancer Research Center (DKFZ), 69120 Heidelberg, Germany; 5National Center for Tumor Diseases (NCT), NCT Heidelberg, a Partnership Between German Cancer Research Center (DKFZ) and Heidelberg University Hospital, 01307 Heidelberg, Germany; 6German Cancer Consortium (DKTK), Core Center Heidelberg, 69120 Heidelberg, Germany; 7Institute of Human Genetics, Heidelberg University, 69120 Heidelberg, Germany; 8Computational Oncology Group, Molecular Precision Oncology Program, National Center for Tumor Diseases (NCT) and German Cancer Research Center (DKFZ), 69120 Heidelberg, Germany; 9Innovation and Service Unit for Bioinformatics and Precision Medicine, German Cancer Research Center (DKFZ), 69120 Heidelberg, Germany; 10Pattern Recognition and Digital Medicine Group, Heidelberg Institute for Stem Cell Technology and Experimental Medicine, 69120 Heidelberg, Germany; 11Department of Radiotherapy & Radiation Oncology, Hubertus Wald Tumorzentrum—University Cancer Center Hamburg (UCCH), University Medical Center Hamburg-Eppendorf, 20251 Hamburg, Germany; 12Department of Translational Medical Oncology, NCT, NCT/University Cancer Center Dresden, a Partnership Between German Cancer Research Center (DKFZ), Faculty of Medicine and University Hospital Carl Gustav Carus, Dresden University of Technology (TUD), and Helmholtz-Zentrum Dresden-Rossendorf, 01307 Dresden, Germany; 13Translational Medical Oncology, Faculty of Medicine and University Hospital Carl Gustav Carus, TUD, 01069 Dresden, Germany; 14German Cancer Consortium (DKTK), Partner Site Dresden, 01307 Dresden, Germany; 15Translational Functional Cancer Genomics, German Cancer Consortium (DKFZ), 69120 Heidelberg, Germany; 16UCCH Kinomics Core Facility, Hubertus Wald Tumorzentrum—University Cancer Center Hamburg (UCCH), University Medical Center Hamburg-Eppendorf, 20251 Hamburg, Germany

**Keywords:** recurrent HNSCC, Trop-2, Sacituzumab govitecan, precision medicine, kinome, tumor models, tumor tissue slices

## Abstract

**Background/Objectives**: In recurrent/metastatic head and neck squamous cell carcinoma (R/M HNSCC), the overall prognosis is poor, and systemic treatment options remain limited. While precision therapy approaches have revolutionized treatment strategies in several tumor types, molecularly informed therapies in R/M HNSCC are rare, primarily due to the low number of actionable genetic alterations identified through next-generation sequencing (NGS) panels. There is an urgent need to establish precision therapy approaches in R/M HNSCC using innovative predictive testing. **Methods**: We report the case of a 43-year-old patient with recurrent oral cancer who was extensively pretreated and comprehensively characterized using both descriptive and functional testing. **Results**: NGS revealed no targetable alterations. A tumor tissue slice radiosensitivity assay suggested radioresistance, arguing against re-irradiation. Kinome profiling identified upregulated Src-family kinases (SFK), and SFK inhibition reduced kinase activity in vitro. Most notably, mRNA analysis demonstrated high Trop-2 overexpression, confirmed by immunohistochemistry (3+ in 100% of tumor cells). Following six cycles of the Trop-2-directed antibody–drug conjugate Sacituzumab govitecan (SG), the patient had an impressive clinical response. **Conclusions**: Tumor characterization beyond genetic profiling can identify novel treatment options in therapy-refractory HNSCC. This is the first report of “real-world” data on promising antitumor efficacy of SG in a heavily pretreated oral cancer patient with Trop-2 overexpression. Consistent with the findings of the Basket TROPiCS-03 study, SG appears to be a promising novel therapy option for R/M HNSCC after failure of immunotherapy and chemotherapy, particularly in patients with Trop-2 overexpression.

## 1. Introduction

Precision oncology focuses on tailoring cancer treatment based on the genetic and molecular characteristics of a patient’s tumor. Comprehensive genomic profiling by next-generation sequencing (NGS) promises to unravel the complex genomic landscape of tumors, offering insights into the ever-expanding spectrum of potential therapeutic targets [1]. In fact, the number of drug approvals based on specific genetic alterations has risen tremendously in recent years [2] and revolutionized treatment concepts in several tumor entities. As an example, genetic analysis has become an established part in the standard diagnostic workflow of lung squamous cell cancer, and BRAF- and MEK-inhibitors dabrafenib and trametinib are approved as palliative first-line therapy in BRAF-mutated cancer [3]. In contrast, head and neck cancers have lagged behind other cancers, with cetuximab, an antibody against EGFR, as currently the only approved targeted therapy option. It is administered without molecular stratification, more regularly in combination with chemotherapy-doublets [4,5], mostly as a palliative second-line treatment line after immunotherapy failure. Molecular analysis via NGS is conducted in special head and neck cancer cancers with rare histopathologies (e.g., salivary gland cancer) or in recurrent or metastatic head and neck squamous cell cancer (R/M HNSCC) patients running out of treatment options. Given that NGS panels only rarely detect targetable lesions in R/M HNSCC and the genomic profile is not predictive for treatment success, innovative technological advancements are needed to delve deeper into the intricacies of the underlying tumor biology. One strategy is to identify oncologic vulnerabilities beyond the DNA level; for example, with mRNA sequencing. Complementary strategies designed to address the limitations of genomics-based precision oncology involve functional testing: Tumor cells from patients are directly challenged with therapeutic approaches to provide immediately translatable, response data to guide therapy.

Herein, we report a case of a 43-year-old patient who presented with aggressive oral cancer with lymph node metastasis and whose tumor biology was analyzed through descriptive and functional precision oncology techniques (genome analysis, mRNA sequencing, kinome analysis, tumor tissue slice radiosensitivity assay) to inform a treatment decision in the molecular tumor board.

## 2. Materials and Methods

The patient’s complex medical history is summarized in Appendix A. In brief, a 43-year-old male patient was diagnosed with cT2 cN0 cM0 oral cancer in 2019. After curative-intent therapy (surgery with reconstruction and adjuvant radiotherapy), the patient faced rapid locoregional and skin-infiltrating recurrence requiring multiple lines of palliative systemic therapy, starting in May 2020. Running out of standard therapeutic options in 2022, intensive functional and molecular characterization was initiated (see Figure 1) in order to identify alterations attackable by targeted therapies.

A fresh tumor specimen was taken from the right outer ear canal after tumor progression in March 2022 for the purpose to undergo the following tests.

### 2.1. Genetic Analysis

Five hundred genes were analyzed using TruSight Oncology 500 Kit from Illumina, San Diego, CA, USA (ISO500 [1]): The sample preparation was carried out according to the manufacturer’s instructions.

### 2.2. mRNA Sequencing

mRNA sequencing was performed as described previously within the DKFZ/NCT/DKTK MASTER (Molecularly Aided Stratification for Tumor Eradication) program [1]. This is a central platform for multidimensional characterization of patients with advanced rare cancers in Germany.

### 2.3. Functional Kinome Profiling

To identify hyperactivated tyrosine kinases and therefore potential targets for molecular therapy, we performed functional kinome profiling as described elsewhere [6,7]. In brief, we compared the activity of phosphorylated tyrosine kinases in fresh-frozen tumor with corresponding normal tissue specimen (PamTechnology^®^, PamGene Int. B.V., s-Hertogenbosch, The Netherlands) (Appendix A Figure A2A–C).

### 2.4. Ex Vivo Tumor Tissue Slice Cultivation and Radiosensitivity Assay

Thin tissue slices precisely cut from a live tumor sample resected from the patient were cultivated and treated with radiotherapy. Radiation-induced DNA damage was quantified by immunofluorescence microscopy for p63/53BP1, as a surrogate for intrinsic tumor radiosensitivity [8,9,10,11]. A standardized protocol is described elsewhere [10,11].

### 2.5. Ethics

The patient gave written and informed consent to be enrolled in the MASTER program (Molecularly Aided Stratification for Tumor Eradication Research); a prospective, continuously recruiting, multicenter observational study [12]. MASTER focuses on biology-guided stratification of oncology treatment, through comprehensive molecular profiling and multidisciplinary tumor board (MTB) decision-making. The study is conducted in accordance with the Declaration of Helsinki, and its protocol (S-206/2011) was approved by the Ethics Committee of the Medical Faculty at Heidelberg University. Comprehensive kinomic testing and an ex vivo radiation sensitivity assay were performed at the University Medical Center Hamburg-Eppendorf. The patient provided written informed consent for the use of excised tumor tissue for analysis. The collection of head and neck tumor tissue was conducted within the framework of the ENT biobank and was duly notified to the Hamburg Commissioner for Data Protection and Freedom of Information (HmbBfDI), in accordance with local laws (§12 HmbKHG), and approved by the local ethics committee (Ethics Commission Hamburg, WF-049/09). Clinical data analysis was performed in compliance with patient consent and local regulations (§7, §8, §12 HmbKHG).

## 3. Results

### 3.1. Results of Descriptive and Functional Testing

○Genetic analysisNo targetable genetic mutations were identified while the tumor mutational burden was moderate (5.2 mutations/MB).○Functional kinome profilingWe identified several kinases of the Src-family (SFK) to be more active in the tumor compared to the normal tissue, representing potential targets (Appendix A, Figure A2). A subsequent in vitro inhibition of SFK using saracatinib revealed a clear inhibition in overall as well as SFK-specific kinase activity in the tumor sample (Appendix A, Figure A2 and Figure A3). The use of alternative TKIs as negative control showed no noteworthy effect.○Ex vivo tumor tissue slice cultivation and radiosensitivity assayQuantitative analysis of DNA double-strand marker 53BP1 in patient-derived tumor tissue slices revealed a low level of DNA damage 24h after radiation (Appendix A, Figure A4).○mRNA sequencingMessenger RNA (mRNA) is transcribed from the DNA sequence of genes and then translated into proteins. Therefore, sequencing mRNA is considered as a more direct measure of the effect of DNA changes on protein products [2]. RNA analysis yielded two relevant results: 117-fold overexpression of Trop-2 (also called TACSTD-2) and 8-fold overexpression of PVRL4 (also called Nectin).

### 3.2. Molecular Tumor Board

A molecular tumor board is a multidisciplinary panel of physicians, clinician scientists, and medical scientists that analyzes a patient’s tumor molecular profile to provide personalized treatment recommendations. In this case, report, techniques for molecular tumor analysis beyond NGS were applied, partly performed at the University Medical Center Hamburg Eppendorf and partly at the DKFZ in Heidelberg in the context of the MASTER program, both sites’ molecular tumor boards being involved in the analysis and treatment recommendation process for this heavily pretreated patient.

Gene sequencing revealed no targetable alterations and ex vivo functional testing of DNA damage repair capacity demonstrated a more radioresistant tumor, arguing against re-irradiation. Functional kinome profiling identified kinases of the Src family (SFK) to be upregulated in the tumor. SFK inhibition showed clear inhibition of kinase activity in vitro, arguing for SFK to be a promising therapy option. This recommendation has only a low level of evidence though, due to the lack of clinical data for the effectiveness of SFK inhibition in HNSCC. In addition, selective SFK inhibitors are not approved for treatment in other tumor entities so far and the predictive value of functional kinome profiling is still a point of current research. mRNA overexpression of Trop-2/TACSTD-2 (117-fold) and PVRL4/Nectin (8-fold) suggests targeted therapy options with higher level of evidence (m2A). There are novel antibody–drug conjugates attacking Trop-2 and PVRL4 already approved for other entities. Trop-2 targeted therapy with Sacituzumab govitecan (SG) is FDA and EMA-approved for the palliative treatment of triple-negative breast cancer patients and urothelial cancer [13]. Enfortumab vedotin is a nectin-4-directed antibody and microtubule inhibitor conjugate, FDA and EMA-approved for the palliative treatment of urothelial carcinoma [14]. The MTB decided for Trop-2 mRNA overexpression to be validated with Trop-2 immunohistochemistry.

Trop-2 staining was strong in all tumor cells (3+ in 100% of tumor cells) (Figure 2). Therefore, the MTB recommended Trop-2 therapy as a potential effective therapy option. The patient’s health insurance agreed to cover the costs for the treatment for 3 months.

### 3.3. Treatment with Trop-2 Targeted Therapy

The patient, in the meantime in ECOG 2 status, received the initial dose of intravenous Sacituzumab govitecan (SG) as per the prescribed regimen (10 mg/kg) on days 1 and 8 of 21-day cycles in our outpatient clinic. After the first dose, there was a discernible halt in tumor growth. Subsequent to the second dose, we observed a consistent regression of the tumor and a remarkable reduction in the malodorous secretion emanating from the tumor mass. The patient tolerated the therapy well with fatigue grade II as sole side effect. By the completion of three cycles (six doses) of SG treatment, a striking clinical response was evident in this heavily pretreated patient (Figure 3). After spending Christmas and New Year’s Eve at home with his family, he decided to stop therapy and died 1.5 month later despite ongoing remarkable tumor control.

## 4. Discussion

Here, we present a case of a 43-year-old patient who suffered from aggressive oral cancer with lymph node metastasis and facial skin infiltration requiring multiple lines of palliative systemic treatment regimens and eventually running out of therapeutic options. In this work, we show the feasibility of using innovative tools to characterize a patient’s tumor biology and transfer the results in a personalized treatment concept in the context of the MTB. To our knowledge, this is the first report that demonstrates promising therapeutic efficacy of Trop-2 directed antibody and topoisomerase inhibitor drug conjugate SG in a heavily pretreated patient based on preclinical molecular stratification.

### 4.1. Role of the MTB for Head and Neck Cancer Therapy

MTBs have an increasingly important role in optimizing treatment by reviewing and interpreting molecular-profiling data [12]. Worldwide, MTBs differ in terms of scope, composition, methods, and recommendations. In Germany, there are 21 centers of personalized medicine forming the German Network for Personalized Medicine (DNPM), amongst them the University Medical Center Hamburg Eppendorf and the University Medical Heidelberg. Recurrent and advanced stage tumor patients are included in the MTB when fulfilling the following characteristics according to DNPM [15]: -Patients without further guideline-based therapy options-Patients with no prospect of success through guideline-based therapies-Patients with rare tumor diseases and no available guideline-based therapy options.

Overall, this is largely consistent with the indications used by other MTB (-networks) [16] and similar to the criteria of the DKFZ/NCT/DKTK MASTER program (Zitat, Ergänzungen). R/M HNSCC progressing under standard systemic therapy are common, but patients do not often present with characteristic targetable lesions, at least when applying genetic testing via next generation sequencing [17]. In line with this, the patient in this case report did not exhibit any targetable genetic alteration, underlining the need for strategies beyond DNA level to detect tumor vulnerabilities.

### 4.2. New Techniques for Precision Therapy in Recurrent and Metastatic Head and Neck Cancer (R/M HNSCC)

In recent years, it has become clearer that a broader spectrum and complementary use of descriptive and functional tests can reveal treatment options and guide therapy decisions in complex or therapy-refractory cases. In the presented case, comprehensive characterization of tumor biology through mRNA sequencing and kinome profiling revealed a number of therapeutic options, specifically Trop-2/PVRL4 or SRC inhibition. Case reports such as this are essential for gaining experience with descriptive and functional tests, which help in developing effective translational strategies to improve treatment outcomes for patients with recurrent/metastatic head and neck squamous cell carcinoma (R/M HNSCC), a particularly challenging condition to treat. This is also important in the context of open questions such as patient selection, technique standardization, cost-effectiveness, and ethical considerations when utilizing personalized therapy.

### 4.3. Trop-2: A Novel Target in R/M HNSCC?

In our report, a heavily pretreated patient showed a tremendous response to SG. Trophoblast cell surface antigen-2 (Trop-2) is a glycoprotein that was first described as a membrane marker of trophoblast cells and was associated with regenerative abilities. Even though its role as an oncogene is currently debated, recent data suggest that Trop-2 acts both as a tumor promoter and tumor suppressor being involved in a set of oncogenetic pathways [18]. Trop-2 overexpression was described in several tumor types; for example, breast, gastric, and ovarian cancers to be associated with accelerated tumor growth and a dismal prognosis. In other tumors like non-small cell lung cancer (NSCLC), metastasis and recurrence are related to Trop-2 downregulation and internalization into the cytoplasm [19]. High Trop-2 levels were expected to identify cancers that are sensitive to Trop-2 targeting therapies [20]. Sacituzumab govitecan (SG) is an antibody–drug conjugate (ADC). ADCs are typically composed of a monoclonal antibody (mAbs) covalently attached to a cytotoxic drug via a chemical linker. Being both a Trop-2 directed antibody and topoisomerase inhibitor drug conjugate, SG was classified as a first-in-class-medication and is EMA and FDA-approved for the second/third-line treatment of metastatic triple-negative breast cancer and metastatic urothelial cancer (regimen (10 mg/kg) on days 1 and 8 of 21-day cycles). Approval for TNBC is based on the phase III study ASCENT [21], which showed a higher progression-free survival (PFS) and overall survival (OS) with SG compared to chemotherapy (5.6 vs. 1.7 months and 12.1 vs. 6.7, respectively) in a heavily pretreated cohort. The overall response rate was 35%. The data from the phase II TROPHY-U-01 trial led to approval of SG in metastatic urothelial cancer after showing an overall response rate of 27% (95% CI, 19.5 to 36.6), with a complete response rate of 5.4%. Eighty percent of patients were pretreated with at least two lines of treatment [22,23]. Currently, several clinical trials with SG are underway, primarily focusing on breast and lung cancer [18].

Preclinical data underline a possible importance of Trop-2 as a target in head and neck cancer: Trop-2 overexpression correlated with lymph node metastasis and dismal prognosis in laryngeal cancer [24]. In 2008, Fong et al. already proposed Trop-2 as a novel prognostic marker for oral squamous cell cancer (OSCC), after analyzing Trop-2 immunohistochemistry expression in a tissue microarray of 90 patients [25]. Zhang et al. confirmed these data when showing that Trop-2 overexpression was higher in poorly differentiated OSCC samples with dismal prognosis. Erber et al. [26] could show that OSCC patients overexpressing Trop-2 had a significant lower 5-year overall survival and recurrence-free survival than those with lower Trop-2 expression, adjusting co-variables (5-year overall survival: 41.2% vs. 55.6%).

Based on the PubMed database, no published case reports exist of a patient treated with SG for head and neck cancer. In March 2025—three years after we treated our patient with SG—the results of the open-label, multicohort Phase II Basket TROPiCS-03 trial were published, underscoring the significance of SG in head and neck cancer (ClinicalTrials.gov identifier: NCT03964727) [27]. Among the 43 HNSCC patients enrolled (including nine with oral cancer), 68% had received ≥2 lines of systemic therapy in the incurable setting. 51% of all patients (22/43) exhibited a reduction in tumor size, with seven patients having a reduction of >30%. The primary endpoint, the objective response rate, was 16% [95% confidence interval (CI), 7–31%]. The clinical benefit rate was 28% (95% CI, 15–44%). The median OS was 9.0 (95% CI, 7.1–10.5) months. The overall survival (OS) rates at 6 and 12 months were 75% (95% CI, 59–86%) and 28% (95% CI, 13–45%), respectively. The wide confidence intervals observed across nearly all endpoints indicate limited precision and reliability of the results, likely attributable to the small sample size. A more in-depth analysis is warranted to confirm the substantial variability in response patterns, which may further underscore the need for predictive biomarkers. The median duration of response was relatively short at 4.1 months. Since our patient passed away four months after SG initiation due to causes unrelated to tumor progression, we were unable to assess the duration of response in this real-world case.

In the TROPiCS-03 trial, the most common treatment-emergent adverse events (TEAE) were diarrhea (47%), nausea (47%), and neutropenia (47%). Grade ≥ 3 TEAE occurred in 58% of patients. In contrast, our patient, despite having an ECOG 2 status, tolerated the therapy well.

Importantly, Trop-2 expression was not analyzed in TROPiCs-03. It may be hypothesized that response rates to Trop-2-targeted therapy in head and neck cancer could be enhanced by selecting patients with high Trop-2 expression. Another area warranting further investigation is whether treatment responses vary among different head and neck cancer sublocations, given that most preclinical data on Trop-2 overexpression were on oral cancer [26].

As of now, no treatment options exist for R/M HNSCC after immunotherapy and chemotherapy failure, highlighting the significance of the basket trial TROP-iCS-03 and our case report, which presents SG as a novel treatment option for these patients.

## 5. Conclusions

Tumor characterization beyond DNA profiling can identify novel treatment options in therapy-refractory HNSCC. This is the first report demonstrating “real-world” data on promising antitumor efficacy of SG in a heavily pretreated oral cancer patient with Trop-2 overexpression. Consistent with the findings of the basket TROPiCS-03 study, SG appears to be a promising novel therapy option for R/M HNSCC after failure of immunotherapy and chemotherapy, particularly in patients with Trop-2 overexpression.

## Figures and Tables

**Figure 1 biomedicines-13-01266-f001:**
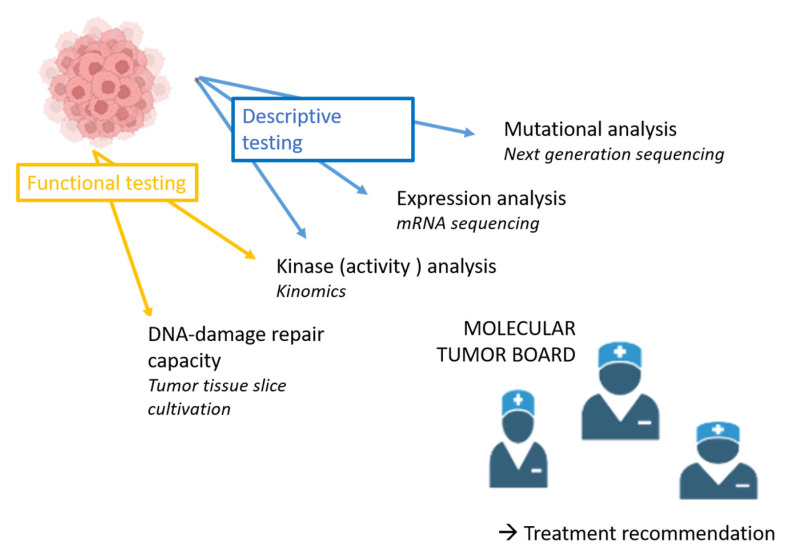
Overview of techniques used for in-depth analysis of the patient’s tumor specimen. Next-generation sequencing and mRNA sequencing are descriptive tests to gain insight into the mutational landscape and the mRNA expression of individual tumors. Functional testing describes how tumor cells are directly perturbed with therapeutic approaches to provide immediately translatable, personalized information to guide therapy. Functional tests used for this patient are (i) radiosensitivity assays of tumor tissue slice cultures (measuring DNA repair capacity) and (ii) kinomics to analyze individual kinome profiles and the effect of specific tyrosine kinase inhibitors.

**Figure 2 biomedicines-13-01266-f002:**
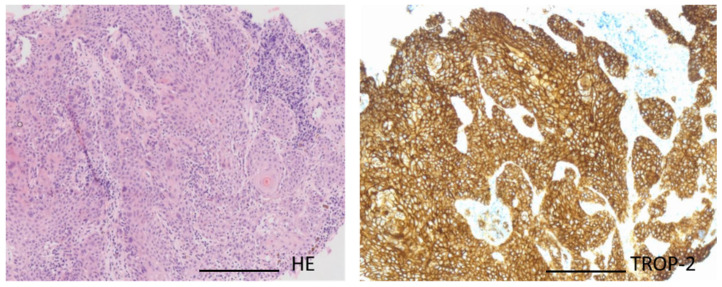
**High Trop-2 expression**. Patient’s tumor specimen stained by hematoxylin–eosin stain (**left**) and Trop-2 (**right**): All tumor cells show strong Trop-2 staining (100% tumor cells 3+) arguing for homogenous Trop-2 overexpression in these tumors and potential effectiveness of Trop-2 targeted therapy. Magnification 40×, scale bar 50 µm. Abbreviations: HE = hematoxylin–eosin staining.

**Figure 3 biomedicines-13-01266-f003:**
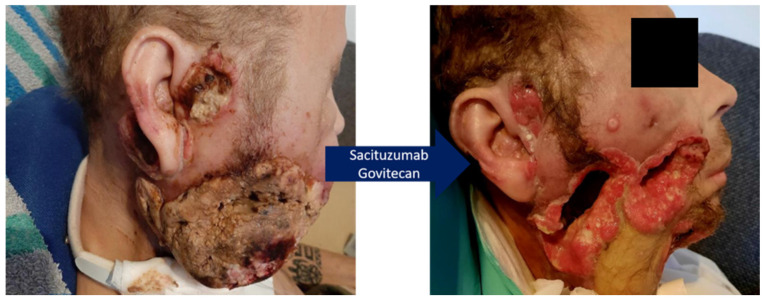
**Rapid tumor response through Sacituzumab govitecan in a patient with recurrent oral cancer on the right side.** Exophytic skin-penetrating tumor mass before (**left**) and after (**right**) 3 cycles of Trop-2 directed antibody–antidrug conjugate Sacituzumab govitecan.

## Data Availability

All research data relevant for this work are available in this manuscript.

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
