# Peer review of "Last Resort? Rationale for Comprehensive Molecular Analysis in Treatment-Refractory R/M HNSCC: A Case Report of Remarkable Response to Sacituzumab Govitecan Following Molecular and Functional Characterization"

_biomedicines, 2025, doi:10.3390/biomedicines13051266_

Round 1
Reviewer 1 Report
Comments and Suggestions for Authors
I enjoyed very much reading your case presentation.
I only suggest some minor editing.. please check lanes:
44 ( over expression), 151 ( two/wo), 165-167 ( rewrite the sentence) “selective SFK-inhibitors, so far, are not approved….., and the predictive value ……, 168 ( indicates ? Suggests? ), 245 ( Sacituzumad Govitecan),252 ( a higher), 266 ( Erber et al [23]); 268 ( than those with lower Trop-2 expression), (Trop-2 expression, adjusting), 269 ( delete 23)
I have a personal comment. Lane 276 and 278 primary endpoint and clinical benefits the confidence intervals 16% (CI) 95%, 7%- 31%), and 28% ( 95% CI, 15%-44%) are quite wide suggesting their findings are not so reliable, this could be due to a very small sample size. This is a important point to keep in mind besides the one you made that you may need the right patient to be identified (Trop-2 expression +)
Comments on the Quality of English Language
I suggested few things to authors. A final review by an editor will be good.
Author Response
I enjoyed very much reading your case presentation.
I only suggest some minor editing.. please check lanes:
44 ( over expression), 151 ( two/wo), 165-167 ( rewrite the sentence) “selective SFK-inhibitors, so far, are not approved….., and the predictive value ……, 168 ( indicates ? Suggests? ), 245 ( Sacituzumad Govitecan),252 ( a higher), 266 ( Erber et al [23]); 268 ( than those with lower Trop-2 expression), (Trop-2 expression, adjusting), 269 ( delete 23)
I have a personal comment. Lane 276 and 278 primary endpoint and clinical benefits the confidence intervals 16% (CI) 95%, 7%- 31%), and 28% ( 95% CI, 15%-44%) are quite wide suggesting their findings are not so reliable, this could be due to a very small sample size. This is a important point to keep in mind besides the one you made that you may need the right patient to be identified (Trop-2 expression +)
Dear Reviewer 1,
Thank you very much for reviewing the manuscript and the constructive response. We made the following corrections:
- Grammar and Writing editing
62: We did not alter the word “overexpression” to “ over expression”, since spelling is considered correct.
178: Correction of “wo” to “two”
195ff: Correction of the sentence:
“This recommendation is based on a low level of evidence: selective SRC family kinase (SFK) inhibitors are not approved for the treatment of any cancer entity. Multi-tyrosine kinase inhibitors with partial activity on the SRC pathway, such as dasatinib—approved for other malignancies—have not been evaluated in HNSCC through clinical trials or case reports. Functional kinome profiling is a relatively novel approach to tumor characterization; however, its translational potential and predictive power has not yet been explored to date.” To
“This recommendation has only a low level of evidence though, due to the lack of clinical data for effectiveness of SFK-inhibition in HNSCC. In addition, selective SFK-inhibitors are not approved for treatment in other tumor entities so far and the predictive value of functional kinome profiling is still a point of current research.”
200: correction of “indicate” to “suggests”
293: correction of “an higher” to “a higher”
307ff: correction of
Erber et al. could show that OSCC patients overexpressing Trop-2 had a significant lower 5-year overall survival and recurrence free survival than these with lower Trop-2 expression,adjusting co-variables (5-year overall survival: 41,2 % vs. 55,6 %) [23].
to
Erber et al. [23] could show that OSCC patients overexpressing Trop-2 had a significant lower 5-year overall survival and recurrence free survival than those with lower Trop-2 expression, adjusting co-variables (5-year overall survival: 41,2 % vs. 55,6 %).
Full text: Consistent terminology for “Sacituzumab govitecan” and “Trop-2”
- The reviewer made a personal comment on the high confidence intervals in the statistical analysis of primary endpoints in the TROPiCS-03 trial study. At his/her point of view this underlines that the results of this study are not so reliable, potentially due to the small sample size. We thank the reviewer for pointing out this aspect and fully agree. We add this point in the text:
320f.: “"The overall survival (OS) rates at 6 and 12 months were 75% (95% CI, 59%–86%) and 28% (95% CI, 13%–45%), respectively. The wide confidence intervals observed across nearly all endpoints indicate limited precision and reliability of the results, likely attributable to the small sample size. A more in-depth analysis is warranted to confirm the substantial variability in response patterns, which may further underscore the need for predictive biomarkers."
Reviewer 1 pointed out that “ The English could be improved to more clearly express the research.”
Therefore, Prof. Kai Rothkamm and Dr. Anna Sophie Hoffmann reviewed and corrected the grammar and language. They were added as co-authors.
Reviewer 2 Report
Comments and Suggestions for Authors
I read this report with a lot of interest, as there are not many examples of incorporating the innovative functional testing of tumor treatment sensitivity in the clinical setting. The case report is clearly presented and well written, and the topic is highly relevant for any clinician working with HNSCC.
There are two points on which I believe the article would benefit from further clarification:
- could you describe in more detail how the Molecular Tumor Board is functioning in your institution? Who are the members? And is which circumstances is it called?
- what is your definition of the radioresistant tumor? I am not aware that your previous publications have defined a cut-off value for that. Has it been established in the meantime?
Author Response
I read this report with a lot of interest, as there are not many examples of incorporating the innovative functional testing of tumor treatment sensitivity in the clinical setting. The case report is clearly presented and well written, and the topic is highly relevant for any clinician working with HNSCC.
There are two points on which I believe the article would benefit from further clarification:
- could you describe in more detail how the Molecular Tumor Board is functioning in your institution? Who are the members? And is which circumstances is it called?
- what is your definition of the radioresistant tumor? I am not aware that your previous publications have defined a cut-off value for that. Has it been established in the meantime?
Dear reviewer 2,
Thank you very much for reviewing our manuscript and for providing valuable suggestions to improve it. We have revised the manuscript in accordance with your comments. Specifically:
- A more detailed description of the Molecular Tumor Board and the associated standards has been added to both the Results and Discussion sections.
183 ff.: A molecular tumor board is a multidisciplinary panel of physicians, clinician scientists and medical scientists that analyzes a patient's tumor molecular profile to provide personalized treatment recommendations. In this case report, techniques for molecular tumor analysis beyond NGS were applied, partly performed at the University Medical Center Hamburg Eppendorf and partly at the DKFZ in Heidelberg in the context of the MASTER (Molecularly Aided Stratification for Tumor Eradication Research) program, both sites’ molecular tumor boards being involved in the analysis and treatment recommendation process for this heavily pretreated patient..
244 ff.: Role of the molecular tumor board for head and neck cancer therapy
MTBs have an increasingly important role in optimizing treatment by reviewing and interpreting molecular-profiling data [12]. In Germany, there are 21 center of personalized medicine forming the German Network for Personalized Medicine (DNPM), amongst them the University Medical Center Hamburg Eppendorf and the National Center for Tumor Diseases (NCT) in Heidelberg. Traditionally, recurrent and advanced stage tumor patients are included in the MTB when fulfilling the following characteristics:
- patients who have already undergone therapy in line with the guidelines
- patients who clinically qualify for molecular-based therapy and
- patients who agree to a possible therapy based on the molecular findings.
R/M HNSCC progressing under standard systemic therapy are common, but patients do not often present with characteristic attackable lesions, at least when applying genetic testing via next generation sequencing [15. This is partly due to the fact that head and neck cancer patients do not often present with characteristic attackable lesions, at least when applying genetic testing via next generation sequencing. In line with this, the patient in this case report did not exhibit any attackable genetic alterations, underlining the need for novel strategies to detect tumor vulnerabilities.
- We addressed your comment: what is your definition of the radioresistant tumor? I am not aware that your previous publications have defined a cut-off value for that. Has it been established in the meantime?
Thank you for bringing this to our attention. We have included additional information in the supplementary image (Figure 4).
Low level of residual radiation-induced DNA damage in the tumor tissue slice assay. Repre-sentative images showing background (top) and residual (bottom) 53BP1 foci (green) following irradiation in patient-derived tumor tissue slices. p63 (red) was used as a tumor marker. After exposure to radiation, there is only a minimal increase in the DNA double-strand break marker 53BP1. A total of 70 tumor cells were analyzed from two biopsies obtained from different tumor regions to consider intratumoral heterogeneity. Quantification revealed 0.2 foci per nucleus 24 hours after irradiation with 3 Gy, with background values subtracted. This value is markedly lower than the median foci count observed in HPV-negative oropharyngeal cancers under the same conditions (1.2 foci per nucleus [11]), suggesting a more radioresistant tumor phenotype.